# Quantitative assessment of colorectal cancer via conditional generative adversarial networks

Quoc Dang Vu[1], Kyungeun Kim[2], Jin Tae Kwak[1,✉]

[1]Department of Computer Science, Sejong University, Seoul, Korea 05006
jkwak@sejong.ac.kr
[2]Department of Pathology, Kangbuk Samsung Hospital, Sungkyunkwan University School of
Medicine, Seoul, Korea 03181

**Abstract.** Grading for cancer, based upon the degree of cancer differentiation, plays a major role in describing the characteristics and behavior of the cancer and determining treatment plan for patients. The grade is determined by a subjective and qualitative assessment of tissues under microscope, which suffers from high inter- and intra-observer variability among pathologists. Digital pathology offers an alternative means to automate the procedure as well as to improve the accuracy and robustness of cancer grading. However, most of such methods tend to mimic or reproduce cancer grade determined by human experts. Herein, we propose a quantitative means of assessing and characterizing cancer via conditional generative adversarial networks. The proposed method is evaluated using tissue microarrays (TMA) of colorectal cancer. The results suggest that the proposed method holds a potential for quantifying cancer characteristics and improving cancer pathology.

**Keywords:** Colorectal cancer, Tumor grading, differentiation, GAN.

## 1    Introduction

In pathology, grading is a means of evaluating tumors based upon their appearance. A grade is given depending on how different the tumors from normal/benign tissues (i.e., differentiation). It is utilized to determine a patient's prognosis and develop a treatment plan. However, pathologists manually assess the tissue under microscope and determine its grade, i.e., it is a subjective and qualitative process, limiting the speed and questioning on the reproducibility [1]. Moreover, it is well known that the current grading system is sub-optimal, especially for prognosis [2] . Therefore, an objective and quantitative method for assessing tumors beyond the current tumor grading scheme holds a great potential for improving cancer pathology and patient management.

  With the advent of digital pathology, numerous computerized tools, including deep learning, have been proposed to aid in pathologists and improve the current pathology [3]. A majority of such tools has been (successfully) applied to discriminative tasks, including cell/tissue classification and segmentation, where the ground truth labels are provided by pathologists. In other words, these tools, by and large, sought to mimic

pathologists and/or automate the histopathologic analysis. Although they could facilitate the rapid and robust decision-making and ease the burden of the pathologists, the limitation of the current histopathologic analysis remains the same. For instance, such tools cannot tell the difference between the tumors within the same grade. To further improve the current grading system and digital pathology tools, it is highly desirable to develop a method that is capable of learning the tissue or tumor characteristics and quantitatively measuring the similarity/dissimilarity to the normal/benign tissue (differentiation) without explicit guidance of tumor grades, i.e., in an unsupervised fashion.

A generative adversarial network (GAN) [4] is a type of deep learning approach that can generate or produce realistic outputs (here, tissue images). Recently, a conditional GAN (cGAN) [5], where the output is conditioned on an input, has gained much attention. For example, a cGAN was used to generate synthetic tissue images [6]. It was also used to conduct virtual H&E staining [7] as well as H&E-to-immunofluorescent stain translation [8]. The strength of a GAN/cGAN is its superior learning capability in an unsupervised manner; hence, the technique could be better suited for learning the latent characteristics of tissues or tumors.

In this manuscript, we propose a cGAN-based method to learn and quantify the characteristics of the tissue that are relevant to tumor differentiation. We construct a cGAN model (BenignGAN) using benign tissue images only. BenignGAN is utilized to generate tumor images of differing degree of differentiation. The less similar the tumor is to the benign (poorly-differentiated), the harder BenignGAN generates a realistic tumor image. The difference between the original and generated tumor images is quantitatively measured and compared to tumor differentiation. We evaluate the proposed method using tissue microarrays (TMA) of colorectal cancer. Our main contributions are summarized as follows: 1) We propose an alternative means of learning and quantifying the tumor characteristics; 2) We build BenignGAN to learn the characteristics of the tissue of origin (here, benign); 3) Employing BenignGAN, the proposed method analyzes tumors in an unsupervised manner, and thus is not restricted to the current grading system.

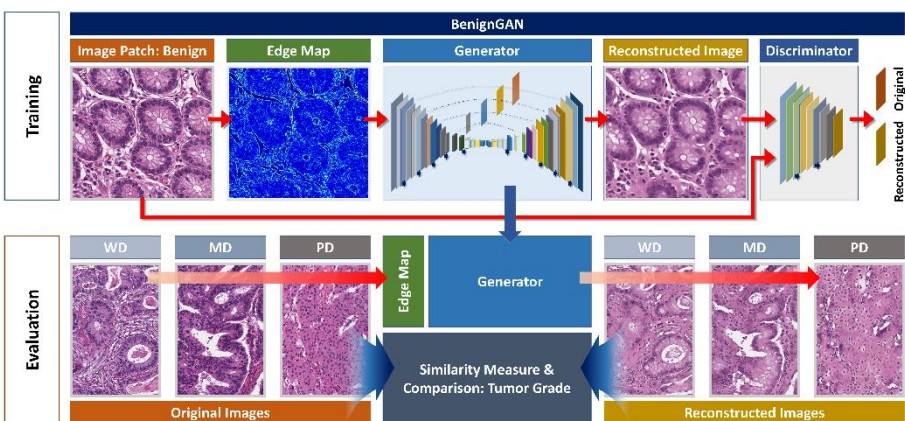

**Fig. 1.** Overview of the proposed method. Benign tissue images are converted to edge maps and used to train a cGAN (BenignGAN). Given the edge map of tumor images of differing grades, BenignGAN is utilized to reconstruct the tumor images. The similarity between the reconstructed and original tumor images is measured and compared to tumor grades provided by pathologists.

## 2 Methods

The overview of the proposed method is illustrated in **Fig. 1**. Details of the method is described in the following sections.

### 2.1 BenignGAN: Conditional Generative Adversarial Network

A conditional generative adversarial network (cGAN) [5] consists of a generator and a discriminator. Given an input image, a generator G learns how to transform the input image to an output image. Following [9], we adopt U-Net [10] architecture to build the generator G. The role of a discriminator D is to distinguish the output images generated by the generator G from the original images. As described in PatchGAN [9], the discriminator D is solely composed of convolutional layers. It outputs a patch, not a scalar. Each pixel in the patch has a value ranging from 0 to 1, representing how believable the corresponding section of the unknown image is.

The overall objective function can be represented as:

$$Loss = arg\ min_G\ max_D\ L_{cGAN}(G, D)\ +\ \lambda\ L_{L1}(G) \tag{1}$$

$$\mathrm{L_{cGAN}}(G, D) = \mathbb{E}_{x,y}[\log(\mathrm{D}(x, y))] + \mathbb{E}_{x,z}[\log(1 - D(x, G(x, z)))] \tag{2}$$

$$L_{L1}(G) = \mathbb{E}_{x,y,z}[\|y - G(x, z)\|_1] \tag{3}$$

where $\mathrm{L_{cGAN}}(G, D)$ the conditional adversarial loss and $L_{L1}(G)$ is the L1 norm loss between the original image and the output image of the generator *G*. *x*, *y* and *z* denote the input image, output image and random noise vector, respectively.

Given an input image *x*, the generator *G* reconstructs the original RGB image *y*. The random noise vector *z* is introduced in the form of dropout to prevent the generator *G* from directly mapping the input image *x* to the output image *y*. L1 norm loss is known to be helpful in generating less blurry output images.

### 2.2 Preprocessing

A cGAN generates an output image conditioned on an input image. Since a neural network tends to focus on the surface statistics of the input [11], training the cGAN directly on the RGB images may cause the generator *G* to only memorize the direct mapping between the input and output and thus fail to learn the fundamental characteristics of the input, i.e., benign tissue. Thus, we limit the amount of information that the generator *G* receives. Given the limited information, the cGAN tries to reconstruct the original RGB image. To reduce the amount of the information, we apply Sobel operator to an input image and compute the gradient magnitude, called as an edge map.

## 2.3    Similarity Metrics

We utilize mutual information (MI), structural similarity index (SSIM) and Pearson correlation coefficient (CC) for measuring the similarity between the reconstructed images and their originals. For each pair of reconstructed and original images, we compute the three metrics for their RGB and gray scale images. For an RGB image, the metrics are separately calculated for each channel and then averaged across channels. Gray scale images are converted from the reconstructed and original RGB images and used to compute the three metrics.

## 2.4    Training and Implementation

We implemented the proposed method using Python and Pytorch. The proposed method is trained for a total of 150 epochs using Adam optimizer with beta1=0.5 and beta2=0.999. $\lambda$ is set to 100 to weight the L1 loss. The learning rate of both the generator and discriminator is set to 1.0e-4 and reduces to 1.0e-5 at 50th epoch. In order to enhance the robustness of the generator, during training, we perform a random horizontal and vertical flip, random scaling, random rotation and random shearing of the input image. We also add Gaussian noise and perform minor blurring using a median or Gaussian filter.

# 3    Experiments

## 3.1    Dataset

One whole slide images (WSI) and two colorectal tissue microarrays (TMAs) were employed to evaluate the proposed method. Tissue samples in the WSI and TMAs were stained with Hematoxylin and Eosin (H&E) and digitized at x40 optical magnification. An experienced pathologist identified and delineated benign and tumor regions. Tumor regions were further categorized into 3 grades – well-differentiate (WD), moderately-differentiate (MD) and poorly-differentiate (PD). From the first TMA group, we extracted 212 benign (BN) image patches of size 1024x1024 and used as the training set. 339 tumor image patches and 80 benign images patches of size 2048x2048 were obtained from the second TMA and WSI, respectively, forming the evaluation set. In short, the evaluation set is composed of 80 BN, 28 WD, 246 MD and 65 PD image patches. The patches were mainly focused on the glandular structure, and the patches containing >20% luminal and/or un-annotated regions were excluded.

## 3.2    Qualitative Results

To qualitatively evaluate the effectiveness of the proposed method, the results of the proposed method is presented in **Fig. 2**. The result demonstrated that BenignGAN is capable of reconstructing the benign tissue image from the corresponding edge map, capturing the underlying characteristics of the benign tissue. The presence and location

of glands, basement membrane and nuclei were well observed and retained. The appearance of glands was also reasonably depicted. However, the reconstructed images had a tendency to become blurry, slightly losing the fine details of the tissue. As for the tumors, the presence of glands and density of nuclei tended to influence the quality of the reconstruction. As the density of nuclei increases, the capability of BenignGAN to reconstruct the original images degrades. Absence of glands in the original image (e.g., PD) resulted in poorer reconstruction.

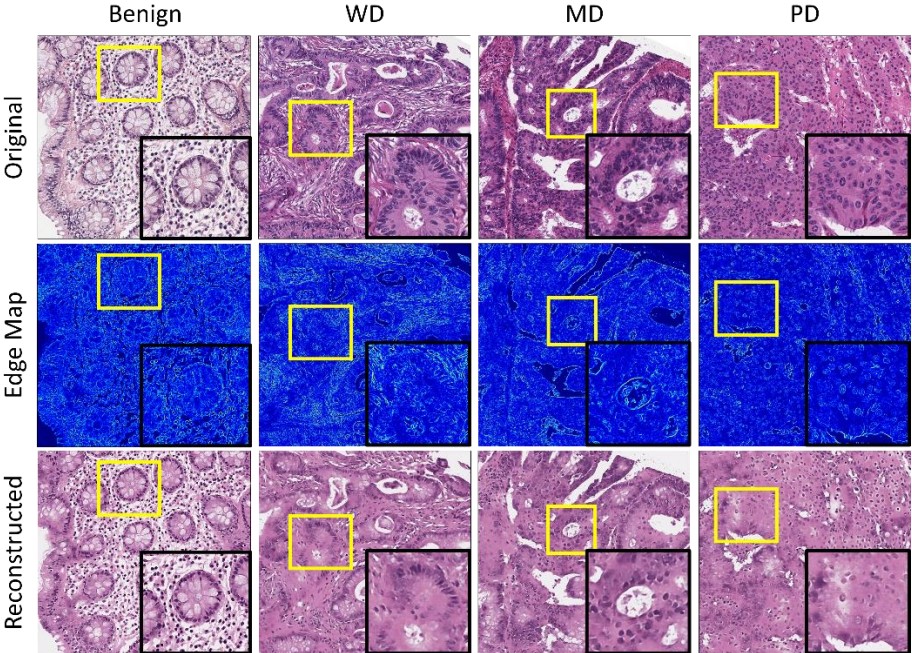

**Fig. 2.** Representative reconstructed and original tumor images. The original tumor images (first row), edge maps (second row) and reconstructed tumor images (third row) are shown for the benign tissue and well-differentiated (WD), moderately-differentiated (MD) and poorly-differentiated (PD) tumor images, respectively.

**Table 1.** Results of the comparison between the reconstructed and original tissue images on the evaluation set. The mean ± standard deviation is shown for three evaluation metrics that is computed for benign and three tumor grades.

|  |  | CC | MI | SSIM |
|---|---|---|---|---|
| **RGB** | Benign | $0.8308 \pm 0.0400$ | $0.6918 \pm 0.0754$ | $0.5130 \pm 0.0385$ |
|  | WD | $0.6544 \pm 0.0802$ | $0.4605 \pm 0.0848$ | $0.4138 \pm 0.0423$ |
|  | MD | $0.6119 \pm 0.0760$ | $0.4369 \pm 0.0768$ | $0.3928 \pm 0.0400$ |
|  | PD | $0.5378 \pm 0.0898$ | $0.3626 \pm 0.0912$ | $0.3520 \pm 0.0415$ |
| **Grayscale** | Benign | $0.8432 \pm 0.0393$ | $0.7470 \pm 0.0806$ | $0.5276 \pm 0.0402$ |
|  | WD | $0.6834 \pm 0.0835$ | $0.4886 \pm 0.0886$ | $0.4191 \pm 0.0442$ |
|  | MD | $0.6414 \pm 0.0809$ | $0.4657 \pm 0.0798$ | $0.3970 \pm 0.0419$ |
|  | PD | $0.5604 \pm 0.0956$ | $0.3922 \pm 0.0950$ | $0.3542 \pm 0.0430$ |

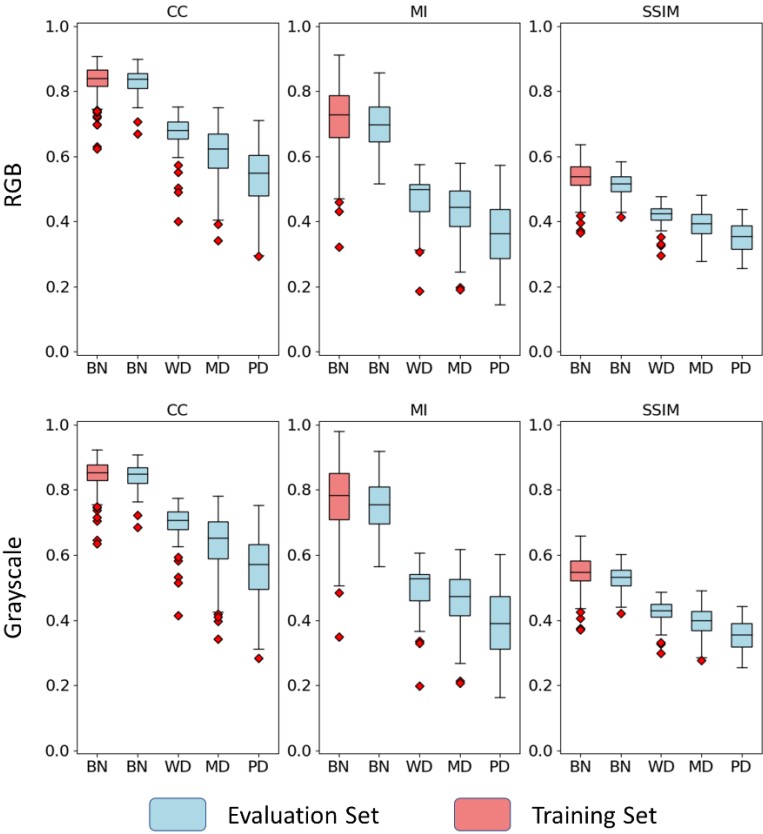

**Fig. 3.** Boxplots for similarity measurements. Correlation coefficient (CC), mutual information (MI) and structural similarity index (SSIM) are measured between the reconstructed and original tumor images, including well-differentiated (WD), moderately-differentiated (MD) and poorly-differentiated (PD) tumors. Red points correspond to outliers, defined by the cases outside the range $[Q1 - 1.5 \times IQR, Q3 + 1.5 \times IQR]$ where Q1, Q3 and IQR denote the first quartile, third quartile and interquartile range, respectively.

### 3.3    Experiments and Quantitative Results

The performance of the proposed method was quantitatively assessed using the three evaluation metrics, including CC, MI and SSIM, between the reconstructed and original tumor images. Within the training set, 3-fold cross validation was performed to evaluate BenignGan's capability on reconstructing benign tissue samples. Comparing the original and reconstructed RGB images, we achieved 0.7984±0.0570 CC, 0.6402±0.1026 MI and 0.4955±0.0491 SSMI. Using grayscale images (converted from RGB images), we obtained $0.8127 \pm 0.0568$ for CC, $0.6920 \pm 0.1104$ for MI and $0.5050 \pm 0.0514$ for SSIM. Subsequently, we trained BenignGAN on the entire training set and tested on the evaluation set. The results are shown in **Fig. 3** and **Table 1**. The results on BN

patches in the evaluation set were similar to the reported results of 3-fold validation (**Table 1**), which confirms the ability of BenignGAN in reconstructing benign tissue images. Moreover, investigation of the results on the tumor image patches revealed that the similarity measurements between the original and reconstructed images are related to the degree of tumor differentiation (**Fig. 3** and **Table 1**). The worse the tumor grade is, the less similar the tumor is to the benign tissue. The similar trend was observed for all three evaluation metrics. ANOVA was further conducted on each of the three evaluation metrics to evaluate the significance of the difference in the similarity measure between the original and reconstructed images in regard to tumor grades. A statistically significant difference (p-value $< 10^{-5}$) was found for the three evaluation metrics using both RGB and grayscale images, suggesting that the difference between the reconstructed and original tumor images, with respect to the benign tissue, could serve as a means of analyzing tumors. In additional, no significant difference between color and grayscale images was observed, indicating that the observed trend is not simply due to the color difference between tumors.

The difference between MD and PD tumors was larger than the difference between WD and MD tumors. This may be ascribable to the presence of glands. Glands are present in many of the tumors of WD and MD but absent in PD tumors. It would have been a bigger challenge for BenignGAN to reconstruct PD tumors since BenignGAN was trained using the benign tissues only, which contain plenty of glands in general. Although there was a downward trend in similarity, the three similarity measures were overlapping between different tumor grades. This may be due to the intrinsic similarity between tumors. However, the specific meanings or biological causes of our observation cannot be identified without further histopathologic and/or biological studies. Moreover, since this study is conducted based upon image patches, sampling of image patches could have an effect on the study. A large-scale study should be followed to further confirm our findings.

## 4    Conclusion

Herein, we presented a method of utilizing a cGAN to quantify the tissue characteristics relevant to tumor differentiation. The experimental results demonstrated that a cGAN is capable of learning the latent representation of the benign tissue and, as it is applied to tumor images, its ability varied depending on the tumor grade, suggesting that it could be utilized to quantitatively analyze and measure the degree of tumor differentiation. The proposed method is generic, and thus could be applied to different types of tissues and tumors. Providing an alternative means of analyzing tissues/tumors, we believe that this approach could aid in improving and reshaping the current cancer pathology in both clinics and research. The future study will entail the comparison of the proposed method to the patients' outcome.

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
