# OpenReview forum: "Quantitative assessment of colorectal cancer via conditional generative adversarial networks"
_MICCAI.org/2019/Workshop/COMPAY — COMPAY 2019_

### Official Review · AnonReviewer2 · 2019-08-12
**Very interesting method for CRC grading. Use reconstruction as a means to access similarity between different grades of tumours and benign. No actual classification. No comparison with standard classification methods.**

**Rating:** 6
**Confidence:** 4

**Review:**

Quality: Well structured.

Clarity: the written is clear and easy to follow.

Originality: the concept is quite interesting. Training cGAN on benign samples and applying it to reconstruct tumor samples. The authors showed that it is more difficult to reconstruct poorly differentiated samples. This makes a lot of sense because the differentiation criterion is based on morphologic similarity the tumor to benign or normal.

Significant of the work: technically interesting but doesn't have a clinical impact because the problem is not clinically relevant.

Pros:
- Interesting way to approach characterizing tumor of different gradings.

Cons:
- the description of the dataset is not straightforward to follow. Perhaps, the authors could include a diagram showing the total number of patients, the total number of samples, the number of samples used in training and testing.
- The more poorly differentiated the tumor, the more difficult the reconstruction. This is okay, but how do I know if a tumor is well, moderately or poorly differentiated. A posterior of belonging to any of these classes would be very helpful.
- The problem with the reconstruction approach is that you need a model of each class to perform reliable classification.
- As the author stated, it is well known that the grading system doesn't provide prognostically relevant information so why not tackle other problem that is more clinically relevant?

---

### Official Review · AnonReviewer4 · 2019-08-15

**Rating:** 6
**Confidence:** 3

**Review:**

The paper is relatively well written and easy to follow. The proposed approach for quantifying tumour tissue is interesting, and it that it was well executed. The main drawback of the paper is the lack of evaluation. While it is clear from the results that there the is a difference between the reconstruction error at the group level, it is not investigated if this can be used to discriminate between the different levels of differentiation.

Furthermore, It is unclear if there is a good separation between training and test cases, which throws some doubt on the experimental setup.  Can patches from the same slide and/or patient be present in both the training and test sets?

The caption of Fig. 1 should appear on the same page as the figure. The input to the discriminator in Fig.1. can be a bit misleading as it appears that both original and reconstructed image are presented at the input at the same time (which I believe is not the case).

Formatting issues:
-	The caption of Fig. 1 should appear on the same page as the figure.
-	Please use the multiplication symbol instead of “x” when specifying resolution and magnification (e.g. “1024×1024” instead of “1024x1024”).

---

### Official Review · AnonReviewer1 · 2019-08-19
**Quantitative assessment of colorectal cancer via conditional generative adversarial networks**

**Rating:** 4
**Confidence:** 5

**Review:**


Claim:  The authors propose a quantitative means of assessing and characterizing cancer via conditional generative adversarial networks : a unsupervised method to learn the latent characteristics, for tumor differentiation, of and quantitatively measuring the similarity to the benign tissue and use to quantify tumor grade. (Benign, WD, MD, PD)..

Clarity/Quality:
1.  cGAN is a known method, used to train on benign tissue images. The authors figures show and claim to reconstruct RGB images. It is claimed that the gradient magnitude of the RGB image is the input (fig 1). It is not clear if  “x” in (eq 1-3) is the RGB image, the gradient image or a combination of both. If RGB alone, not sure how the loss function in (3) is impacted as only the gradient magnitude is the G input?
I am not sure if the with edge image input alone, can GAN reconstruct the underlying color image?
The authors need to clarify, what exactly the input data is - gradient image or the original RGB image or both, and what the reconstruction is ?

2 . Similarity Metrics:  As the original and reconstructed images are of same modality, mutual information metric is redundant relative to Pearson CC metric (as results in Table 1 also show the same trends).

Quality of the Paper:  Poor.
In regards to tumor differentiation, the authors claim the proposed method potentially can reveal the “difference between the tumors within same grade”. But no supporting data is presented to substantiate the claim. The given similarity metrics only reflect the specific image reconstruction quality.  If anything, the result images show the reconstructed images are poor reconstructions of the original images as tumor grade increases, thus any within grade image tumor differences may be lost in the reconstruction process.

The latent characteristics that are captured by the model - may be mainly the glandular appearance in the benign images. The similarity metric decreasing trend from benign to PD may reflection of the increasing disappearance of the overall glandular architecture from benign, WD, MD to PD.

The presented results and images are not conclusive enough to understand what exactly is being captured by the model. It is worth noting that glandular macro-architecture is only one of the components of the clinical scoring guideline. Cellular appearance and nucleoli visibility are other attributes, not clear if and how those attributes are captured by the model.

Completeness:
The paper is incomplete. The proposed method, built upon known methods, is not clear on the novelty.  Only a small dataset is used for training. The results are not adequate to support the claims. (Apart from the similarity metric trend follow the clinical scoring trend).

Given the data set used for training and testing is quite small, not sure if it is possible to arrive at any definitive conclusions.

To improve the paper, I would recommend,
1:  Take a smaller task to show the reconstruction capability for benign grade images alone.

2. To evaluate tumor differentiation claims, use a larger dataset of different tumor grades to patterns in reconstruction errors and similarity scores to be able to evaluate the potential of the reconstructed image similarity metric ( Pearson CC is good enough) as a clinical grading criterion.

---

### Decision · Program_Chairs · 2019-08-20

Accept